# Toxicological Assessment of 2-Hydroxychalcone-Mediated Photodynamic Therapy: Comparative In Vitro and In Vivo Approaches

**DOI:** 10.3390/pharmaceutics16121523

**Published:** 2024-11-26

**Authors:** Níura Madalena Bila, Carolina Orlando Vaso, Jenyffie Araújo Belizário, Letícia Ribeiro Assis, Luís Octávio Regasini, Carla Raquel Fontana, Ana Marisa Fusco-Almeida, Caroline Barcelos Costa-Orlandi, Maria José Soares Mendes-Giannini

**Affiliations:** 1Department of Clinical Analysis, School of Pharmaceutical Sciences, Universidade Estadual Paulista (UNESP), Araraquara 14800-903, SP, Brazil; niura.madalena.bila@gmail.com (N.M.B.); carolovaso@hotmail.com (C.O.V.); jee.abelizario@gmail.com (J.A.B.); carla.fontana@unesp.br (C.R.F.); ana.marisa@unesp.br (A.M.F.-A.); carolbarceloscosta@gmail.com (C.B.C.-O.); 2Department of Public Health, School of Veterinary, Universidade Eduardo Modlane (UEM), Maputo 257, Mozambique; 3Department of Chemistry and Environmental Sciences, Institute of Biosciences, Humanities and Exact Sciences, Universidade Estaudal Paulista (UNESP), São José do Rio Preto 01049-010, SP, Brazil; leticia-rda@hotmail.com (L.R.A.); luis.regasini@unesp.br (L.O.R.)

**Keywords:** 2-hydroxychalcone, photodynamic therapy, monolayers, three-dimensional tissue model, toxicological evaluation of compounds, *Galleria mellonella*, *Caenorhabditis elegans*

## Abstract

Background: Photodynamic therapy (PDT) is a treatment modality that uses light to activate a photosensitizing agent, destroying target cells. The growing awareness of the necessity to reduce or eliminate the use of mammals in research has prompted the search for safer toxicity testing models aligned with the new global guidelines and compliant with the relevant regulations. Objective: The objective of this study was to assess the impact of PDT on alternative models to mammals, including in vitro three-dimensional (3D) cultures and in vivo, in invertebrate animals, utilizing a potent photosensitizer, 2-hydroxychalcone. Methods: Cytotoxicity was assessed in two cellular models: monolayer (2D) and 3D. For this purpose, spheroids of two cell lines, primary dermal fibroblasts (HDFa) and adult human epidermal cell keratinocytes (HaCat), were developed and characterized following criteria on cell viability, shape, diameter, and number of cells. The survival percentages of *Caenorhabditis elegans* and *Galleria mellonella* were evaluated at 1 and 7 days, respectively. Results: The findings indicated that all the assessed platforms are appropriate for investigating PDT toxicity. Furthermore, 2-hydroxychalcone demonstrated low toxicity in the absence of light and when mediated by PDT across a range of in vitro (2D and 3D cultures) and in vivo (invertebrate animal models, including *G. mellonella* and *C. elegans*) models. Conclusion: There was a strong correlation between the in vitro and in vivo tests, with similar toxicity results, particularly in the 3D models and *C. elegans*, where the concentration for 50% viability was approximately 100 µg/mL.

## 1. Introduction

Alternative models for assessing reproductive and developmental toxicity are increasingly aligned with the 3Rs principles (reduction, refinement, and replacement) for using animals in regulatory risk assessment. The 3Rs concept, introduced by Russell and Burch in 1959, advocates using alternative methods, replacing animals, and improving ethical standards in research [1]. Three-dimensional (3D) cell cultures have been widely used since the 20th century in toxicological studies, host–parasite interactions, and degenerative diseases. However, the use of this platform in some areas, such as photodynamic therapy (PDT), has been little explored. This type of culture has some advantages, such as greater cell–cell and cell–matrix interactions, the ability to create an in vitro microarchitecture similar to the in vivo microarchitecture, low cost, ease of production and maintenance, and faster results compared to conventional animals [2,3,4,5].

Another approach to replacing or reducing the use of mammals in scientific research is fusing lower animals, such as invertebrates. These alternative models offer several advantages over mammals, including lower cost, easier maintenance, and fewer ethical concerns [6,7]. The most used invertebrate models are *Artemia salina*, *Drosophila melanogaster*, *Galleria mellonella*, and *Caenorhabditis elegans*. For vertebrates, we have *Danio rerio* (zebrafish) and *Oryzias latipes* (medaka fish) [6,8,9].

*G. mellonella* has been widely used as an experimental model to study microbial virulence, antimicrobial efficacy, and the toxicity of experimental molecules [10,11]. Meanwhile, *C. elegans* has a well-established history as a model organism in genetics, developmental biology, and aging research [12]. In particular, virulence studies and toxicological evaluation of new molecules have increased [13]. In vivo testing provides a comprehensive assessment of pharmacological efficacy and safety by capturing the complex interactions of compounds within a living organism. In addition, the combination of in vitro and in vivo models provides a more thorough evaluation of compounds, allowing for a broader understanding of their effects on specific tissues as well as their overall impact on the animal [6,8,9].

Very few studies have evaluated the toxicity of PDT in alternative in vivo and 3D cell culture models for antimicrobial treatment. Because of the high prevalence of antimicrobial resistance, PDT has recently gained attention as a promising antimicrobial alternative, particularly in combating microbial resistance and eradicating biofilms [14,15]. PDT has been established in microbiology for over 100 years [16]. The mechanism of microbial destruction involves a combination of a photosensitizer, oxygen, and light of a specific wavelength, resulting in oxidative stress and cell death [17,18].

2-hydroxychalcone is a chalcone derivative showing antifungal and antibiofilm activity, specifically against dermatophyte species (fungi that cause skin infections) and *Histoplasma capsulatum* [19,20]. This compound has photosensitizing properties that enhance its effect on planktonic cells and dermatophyte biofilms, fungi highly prevalent in fungal skin diseases [19]. Studies suggest that the likely mechanism of action of 2-hydroxychalcone-mediated PDT involves the generation of reactive oxygen species (ROS), inducing cell death by apoptosis and necrosis. In addition, 2-hydroxychalcone shows tropism for the cell membrane and wall, possibly interacting with cell wall glycoproteins and membrane ergosterol [19].

Human fungal infections can cause more than 1.5 million deaths annually, associated with the emergence of fungal strains that are less susceptible to antifungal drugs or rapidly evolving drug resistance [21,22]. Dermal fungal infections caused by dermatophytes are already considered one of the most difficult diseases to treat, resulting in lengthy treatments and a higher likelihood of recurrences [23]. As a result, there is an ongoing search for compounds that are both effective and have low toxicity. In addition, the need to reduce or eliminate the use of mammals in scientific experiments is a prominent and continually debated issue. Consequently, there is an urgent need to develop safer toxicity test models that comply with emerging global guidelines [24].

Therefore, this study aimed to evaluate the effects of PDT using a potent photosensitizer, 2-hydroxychalcone, in alternative mammalian models. This approach included in vitro assessments in 3D cultures and in vivo evaluations using invertebrate animals.

## 2. Material and Methods

### 2.1. Characterization of Three-Dimensional Cultures of Human Skin Keratinocytes (HaCat) and Human Dermal Fibroblasts (HDFa)

#### 2.1.1. Three-Dimensional Culture Using the Agarose Coating Method

Two cell lines were utilized in this study: primary dermal fibroblasts (HDFa) (ATCC PCS-201-012) and adult human epidermal keratinocytes (HaCat) (CLS Cell Lines Service, 300493). The 3D cell spheroid model was developed as described by Vaso and collaborators [25]. Ninety-six well plates (Kasvi, São José dos Pinhais, Paraná, Brazil) were prepared and coated with 50 µL of agarose at a concentration of 1.5% (LCG). The cells were prepared in concentrations ranging from 1 × 10^3^ to 6 × 10^4^ cells/well in Dulbecco’s modified Eagle’s medium (DMEM) (Gibco^®^, Thermo Fisher Scientific, Waltham, MA, USA) without phenol red, supplemented with 10% of fetal bovine serum (FBS) (Sigma-Aldrich, Milano, Italy). The suspensions were transferred to the wells with agarose. Subsequently, the plates were incubated at 37 °C for 96 h in an atmosphere of 5% CO_2_; this time was necessary to form the spheroid [2].

#### 2.1.2. Diameter Determination

The following parameters were considered to evaluate the spheroids formed at different cell concentrations and in both strains: integrity, shape, diameter, and one spheroid per well. For this purpose, images were taken on the fourth day of spheroid formation using the In Cell Analyzer 2000 (GE Healthcare, Chicago, IL, USA). The diameter of the spheroids was verified by ImageJ 1.53 (Fiji) software, considering the horizontal diameter with values between 370 and 600 µm for the standardization of the model [2,25].

#### 2.1.3. Cell Viability by the Resazurin Method

The metabolic activity of spheroids was measured by the resazurin reduction method. After 96 h of incubation, 20 µL of the 50 µM resazurin solution (Sigma-Aldrich, Milano, Italy) was added to each well containing the spheroids and incubated for 24 h under standard conditions. Absorbance was measured at 570 and 600 nm in a spectrophotometer (Epoch, Biotek, Santa Clara, CA, USA). Monolayer cells (2D) were used as a control [25,26].

### 2.2. 2-Hydroxychalcone Dilution

2-hydroxychalcone was solubilized in 100% dimethyl sulfoxide (DMSO) (Synth, Diadema, Sao Paulo, Brazil). A 30,000 µg/mL stock solution was prepared and stored at −80 °C. The compound was diluted in Roswell Park Memorial Institute (RPMI) 1640 medium (Gibco^®^, Thermo Fisher Scientific, Waltham, MA, USA), without sodium bicarbonate, with phenol red as a pH indicator, and buffered with MOPS of 4-morpholinopropanesulfonic acid hemisodium salt (Sigma-Aldrich, Milano, Italy), pH = 7, so that the DMSO concentration was less than 1% to perform the assays. This compound has already been shown to have antifungal activity and can be enhanced with PDT, as previously demonstrated [19]. The synthesis of 2-chalcone was carried out as described by Melo et al. [21] and Zeraik et al. [27]. The compound was functionalized by the addition of a hydroxyl group at carbon 2 to increase its solubility in aqueous media, and its chemical structure and absorption spectrum are shown in Figure 1.

### 2.3. Cytotoxicity in the Monolayer Model (2D)

The toxicity of 2-hydroxychalcone in the dark mediated by PDT on HDFa and HaCat cells was verified using the resazurin colorimetric method. The cell lines were maintained in cell culture flasks together with DMEM medium containing 10% FSB and without phenol red, which can interfere with cell irradiation, and incubated in an incubator under standard conditions (37 °C, 5% CO_2_). Cell suspensions were prepared to obtain a final concentration of 5 × 10^2^ and 2 × 10^4^ cells per well in the 96-well microdilution plate in a volume of 200 μL for HDFa and HaCat, respectively. After 96 h of incubation, 1 to 500 µg/mL of 2-hydroxychalcone was added to the cells and incubated for 10 min in the dark. Then, some plates were irradiated with blue LED (IrradLED^®^-Biopdi, Sao Carlos, SP, Brazil) with a wavelength range between 455 nm and 492 nm, which was used as the light source. Intensity was maintained at 58 mW/cm^2^, and the administered dose was 150 J/cm^2^ (43 min), while others were kept in the dark. After irradiation, all plates were incubated for 72 h at 37 °C, 5% CO_2_, and protected from light. A total of 20 μL of resazurin reagent was added to the wells, and cell viability was read using a spectrophotometer at wavelengths of 570 and 600 nm [26]. The control was performed without adding 2-hydroxychalcone in two different conditions (plates with and without LED irradiation).

### 2.4. Cytotoxicity in a Three-Dimensional (3D) Model

For the cytotoxicity assays, spheroids were prepared as described in Section 2.1.1. After the incubation period for the formation of the 3D models, 100 µL of DMEM medium was removed, and 100 µL of DMEM containing different concentrations of 2-hydroxychalcone (1 to 500 µg/mL) was added [25]. The plates were then incubated for 10 min and protected from light. Some plates were irradiated with blue LED at 150 J×cm^−2^, while other plates were kept in the dark. After irradiation, the plates were incubated for another 72 h under the same conditions. Twenty microliters of resazurin were added after the incubation period to read cell viability on the spectrophotometer at wavelengths of 570 and 600 nm. [26]. The control was performed without adding 2-hydroxychalcone in two different conditions (plates with and without LED irradiation).

### 2.5. In Vivo Toxicity of 2-Hydroxychalcone in the Dark and Photoexcited

#### 2.5.1. Alternative Model—*C. elegans*

The wild-type N2 and mutant AU37 (glp-4(bn2) I; sek-1(km4) X) strains were employed in the toxicity assay conducted on *C. elegans*. Both were maintained in NGM (Nematode Growth Medium) seeded with Escherichia coli OP50 and incubated at 15 °C following standard protocols [28]. Approximately 24 h after cultivation, synchronization was performed to ensure all larvae were at the L4 stage. The eggs were collected by washing the plates with approximately 5 mL of a 50 mM NaCl solution and transferred to 15 mL conical tubes for sedimentation. Subsequently, the supernatant was removed, and the eggs and larvae were washed at least three additional times to remove *E. coli*. Following the final wash, approximately 5 mL of hypochlorite solution (4 mL of 1M NaOH; 1.3 mL of NaOCl (Sigma-Aldrich, Milano, Italy); 10.7 mL of sterile distilled water) was added and shaken for three minutes, after which centrifugation at 2000 rpm was performed for one minute. The supernatant was promptly removed, and the eggs were washed thrice with 50 mM NaCl solution. A Pasteur pipette was used to transfer the sediment, which contained the eggs and dead adult larvae, to NGM medium plates lacking bacteria. These plates were then incubated at 16 °C for 24 h. After the eggs hatched and the L1 larvae migrated, they were collected by washing the plates with 50 mM NaCl and transferred to NGM medium plates with *E. coli* OP50 until they reached the L4 larval stage approximately three days after synchronization. Approximately 20–30 larvae were transferred to each well of a 96-well plate, which was composed of 100 μL of a medium containing the following components: 60% 50 mM NaCl; 40% BHI broth; 10 mg/mL cholesterol in ethanol; 200 mg/mL ampicillin; 50 mg/mL gentamicin; and 90 mg/mL kanamycin. Subsequently, 100 μL of the 2-hydroxychalcone dilutions were added, resulting in final concentrations ranging from 1.98 to 500 µg/mL in each well. The plates were incubated at 25 °C for 24 h. The survival of the nematodes was assessed based on their mobility and shape (rod-shaped larvae were considered dead, while sinusoidal larvae were considered alive) under a phase-contrast inverted light microscope (Olympus CH30) with 40× objective lenses [28,29]. The control was performed without adding 2-hydroxychalcone in two different conditions (plates with and without irradiation with LED).

#### 2.5.2. Alternative Model—*G. mellonella*

The toxicity assay in *G. mellonella* was conducted following the methodology described by Singulani et al. [30]. Briefly, *G. mellonella* larvae with weights between 0.2 and 0.3 g and without dark spots were selected and maintained in Petri dishes (n = 10 for each group) in the dark at 37 °C overnight before the experiments (Appendix A). Solutions of 2-hydroxychalcone in PBS were prepared to be administered at the following doses: 10, 50, 100, and 200 mg/kg. Ten microliters of the solutions were injected into the last pair of prolegs (always the right proleg) using a Hamilton syringe after the larvae had been previously cleaned with 70% ethanol (Appendix A). A control group of larvae treated with PBS and another without treatment were used as controls in the tests. The viability of the larvae was monitored daily for seven days. The larvae were considered dead when they exhibited no movement in response to touch. The control was performed without treatment with 2-hydroxycalcone in the two distinct conditions (larvae with and without irradiation with LED) (Appendix A).

### 2.6. Statistical Analysis

In the in vitro assays, the CC_50_ was calculated using a non-logarithmic semi-linear model. The alternative models of *Galleria mellonella* were plotted using the Kaplan–Meier method and subsequently analyzed using the Log-rank (Mantel–Cox) test. All data were subjected to statistical analysis using one-way or two-way analysis of variance (ANOVA) with the Bonferroni post-test, employing GraphPad Prism 5.0 software. Three tests were conducted, and the resulting data were subjected to statistical analysis, with the mean and standard deviation being calculated. A *p*-value of less than 0.05 was considered statistically significant.

## 3. Results

### 3.1. Characterization of Three-Dimensional Models of Dermal Cells—Diameter and Viability

The ideal spheroid was defined as one that exhibited a circular shape, with intact cell–cell and cell–matrix connections, an external membrane without ruptures, and a diameter of approximately 370 to 600 µm. The optimal concentrations identified were 5 × 10^2^ cells/well for the HDFa line and 2 × 10^4^ cells/well for the HaCat line. The mean diameter of the spheroids derived from the HDFa line was 542.07 ± 18.15 µm, while those derived from the HaCat line had a mean diameter of 399.49 ± 45.53 µm (Figure 2A). The cell viability tests demonstrated that both 3D models exhibited over 94% viability by the fourth day of formation (Figure 2B). The HDFa line exhibited an average viability of 94.54%, while the HaCat line demonstrated 99.25% viability. These results indicate that both cell lines have an optimal cell concentration that allows sufficient oxygenation and cell–cell interactions.

Images were captured on the fourth day of culture to assess spheroid integrity and confirm the formation of a single spheroid per well. The images of both HDFa and HaCat spheroids demonstrate well-formed, complete, and rounded structures with only a single spheroid present per well (Figure 3).

### 3.2. Evaluation of the Cytotoxicity of 2-Hydroxychalcone Mediated by Photodynamic Therapy (PDT) and in Dark Conditions, Using Monolayers and a Three-Dimensional Model with the HDFa and HaCat Cell Lines

The results of the cytotoxicity of 2-hydroxychalcone and 2-hydroxychalcone mediated by PDT in a 2D and 3D model of the HDFa cell line are presented in Figure 4. No differences were observed in the toxicity of cells maintained in the dark and exposed to blue light in each cell model (2D and 3D). Furthermore, the 3D model presented cytotoxic concentration values that inhibit 50% of cell proliferation (CC_50_) of 33.57 µg/mL and 36.71 µg/mL in cells protected from light, in the 2D and 3D models, respectively. The irradiated cells showed a CC_50_ of 100 µg/mL and 100.8 µg/mL in the 2D and spheroid models, respectively. Photoexcited 2-hydroxychalcone conferred lower toxicity to cells when compared with light-protected 2-hydroxychalcone in both models (*p* < 0.001).

Regarding the results of HaCat cells (Figure 5), it was demonstrated that this cell line exhibited heightened sensitivity to photoexcited 2-hydroxychalcone compared to the same compound devoid of light excitation across both models. Additionally, 2-hydroxychalcone exhibited lower toxicity in the three-dimensional model compared to the 2D model under both photoexcited and dark conditions. The CC_50_ values for light-exposed cells were 66.58 µg/mL in the 2D model and 104.90 µg/mL in the three-dimensional cultures. In the absence of light, the CC_50_ was 84.16 µg/mL in the 2D model and 152.0 µg/mL in the spheroid model.

### 3.3. In Vivo Toxicity Assay

#### 3.3.1. Acute Toxicity in *Caenorhabditis elegans*

The acute toxicity of 2-hydroxychalcone in both dark and photoexcited conditions was evaluated in *C. elegans* at the L4 larval stage (L4). Figure 6A illustrates the survival percentage of the AU37 mutant strain larvae after contact with different compound concentrations in both dark and photoexcited conditions. It was observed that as the concentration of the compound increased, the survival rate of the larvae decreased. Larval mortality reached approximately 30% at 125 µg/mL or higher concentrations without light. Regarding the photoexcited 2-hydroxychalcone, the degree of toxicity was markedly elevated compared to the dark treatment, with a larval mortality rate of approximately 52% (*p* < 0.001) at a concentration of 125 µg/mL. The wild-type strain (N2) was more resistant to treatment but within the test error of the mutant strain (Figure 6B). However, an identical toxicity profile was evident at concentrations of 125 µg/mL, with only 21% and 42% of the larvae dying after treatment with 2-hydroxychalcone in dark and photoexcited conditions, respectively.

Figure 7 and Figure 8 illustrate the effects of 2-hydroxychalcone treatment on the larvae of strain N2. The figures depict the results of treatment in the dark (Figure 7) and under photoexcitation (Figure 8). In both strains, the highest concentrations of the compound resulted in the observation of dead larvae (characterized by an absence of movement and a rod-shaped morphology) and a few live larvae. At the lowest concentrations of the compound, larvae with a sinusoidal shape and/or movement (i.e., live larvae) were observed.

#### 3.3.2. Toxicity in *G. mellonella*

The in vivo toxicity of 2-hydroxychalcone in the dark and photoexcited 2-hydroxychalcone was evaluated using *G. mellonella* larvae. For this purpose, different doses of 2-hydroxychalcone were evaluated in the dark: 10, 50, 100, and 200 mg/kg, and the same concentrations were used for the photoexcited compound. No significant toxicity was observed in the larvae at all doses and conditions tested. Even at the highest concentrations (200 mg/kg), over 90% of larvae survived after seven days (Figure 9). The control larvae (treated with PBS, untreated, and irradiated only with blue LED) also exhibited survival rates exceeding 90% until the seventh day (Figure 9).

## 4. Discussion

PDT is a promising approach with notable effectiveness against several species of microorganisms, including those resistant to conventional drugs [31,32]. PDT generally has a favorable safety profile, with minimal side effects and complications compared to other treatment methods [33]. With the advent of tissue engineering, the combination of PDT and three-dimensional models has gained increasing attention, as these cultures provide an excellent platform for studying the cytotoxicity of bioactive compounds with therapeutic potential. Cellular responses to PDT have been observed to differ between two-dimensional and three-dimensional culture models. In 2D cell cultures, all cells receive a uniform dosage of the photosensitizer and oxygen. On the other hand, in three-dimensional models, the photosensitizer and oxygen must diffuse to the spheroid’s center, leading to reduced levels of these compounds in the central region [34].

One of the most utilized parameters for the assessment of spheroids is their diameter, which helps assess how accurately 3D cultures replicate the in vivo environment. Spheroids smaller than 200 μm or above 600 μm may face limitations in the diffusion of molecules and oxygen [2,35,36,37]. In this study, the spheroids presented average diameters, comparable to those in the literature, with values of 542.07 µm for the HDFa line and 399.49 µm for HaCat line in cell concentrations of 5 × 10^2^ cells/well for the HDFa line and 2 × 10^4^ cells/well for the HaCat line. Furthermore, the spheroids exhibited high cell viability, exceeding 94%, with a well-defined spheroidal structure, intact membrane, and interconnected cells. This organization supports efficient oxygen distribution, promoting a high proliferation rate and robust development. The suitability of these parameters enabled the cultures to be used for more realistic cytotoxicity tests, which reflect cellular interactions and treatment responses. Chen et al. [34] determined that spheroids with diameters ranging from 250 to 450 µm were optimal for conducting cytotoxicity assays using PDT. One of the fields that extensively uses phototherapy is dermatology. As a result, the use of dermal cell lines has emerged as a promising area of study, gaining increasing attention, especially with the widespread ban on the use of vertebrate and invertebrate animals for cosmetic testing in many countries, aligning with the cruelty-free movement [38,39].

The findings of our study indicate that three-dimensional dermal cell models are viable and potential tools for assessing the toxicity of compounds kept in the dark and mediated by PDT. Furthermore, they showed that the three-dimensional model is less sensitive to toxicity from 2-hydroxychalcone when compared to the 2D model. Previous studies have also reported that 3D models exhibit lower toxicity than two-dimensional models (2D). This sensitivity of cultures may be attributed to varying levels of perfusion of photosensitizers, oxygen, and the expression of genes and proteins [25,34,40,41].

Our findings showed that PDT was more toxic to the HaCat line than the HDFa line. Keratinocytes are the most abundant cell type found in the epidermis. They are characterized by their high keratin content, enhancing the skin’s resistance to external stresses and impermeability. In contrast, fibroblasts are in the deeper dermis layer, which plays a crucial role in maintaining the structural integrity of connective tissue and synthesizing proteins. However, fibroblasts are less resistant to external factors than keratinocytes [42].

Studies aimed at classifying toxicity in *C. elegans* have consistently demonstrated a strong correlation with oral LD_50_ classifications in rodents, highlighting the model’s effectiveness for screening molecules with biological activity. In contrast to mammalian studies, a *C. elegans* larval toxicity assay can be performed by a single technician in a shorter timeframe using essential equipment [43]. Despite the extensive research on the toxicological evaluation of new and conventional compounds in *C. elegans* larvae, the toxicity of PDT in larvae has not yet been reported [29,30,44]. The results of this study indicate that *C. elegans* is a promising model for the toxicological evaluation of PDT, demonstrating that larval viability is not impacted by light exposure (in the absence of a photosensitizer), even at very high energy doses (150 J/cm^2^). Furthermore, it was found that photoexcited 2-hydroxychalcone conferred slightly more significant toxicity compared to 2-hydroxychalcone in the dark.

Recently, scientists have initiated trials to assess the viability of *G. mellonella* in PDT toxicological evaluation [45,46]. This alternative model can detect the toxic effect of the compound earlier, saving time and money while reducing the number of mammals needed [28]. Marras et al. [46] found that the survival of *G. melonella* larvae was above 90% in larvae subjected to PDT, such as those kept in the dark. Our findings present similar results, showing that 2-hydroxychalcone does not confer toxicity to this experimental model at the concentrations tested.

## 5. Conclusions

In contemporary research, invertebrate models are effectively utilized to preliminarily evaluate the toxicity of new compounds. Based on the findings presented in this study, it can be concluded that all models employed are suitable for assessing the toxicity of PDT. An excellent correlation was also observed between in vitro and in vivo tests, with similar toxicity results, particularly in the 3D models and *C. elegans*, where 50% viability concentrations were observed at approximately 100 µg/mL.

## Figures and Tables

**Figure 1 pharmaceutics-16-01523-f001:**
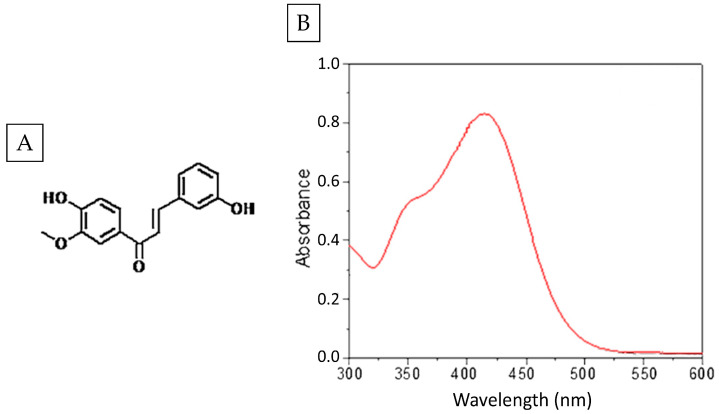
Chemical structure of 2-hydroxychalcone (**A**) and absorption spectrum (**B**). Adapted from [20], Elsevier, 2017.

**Figure 2 pharmaceutics-16-01523-f002:**
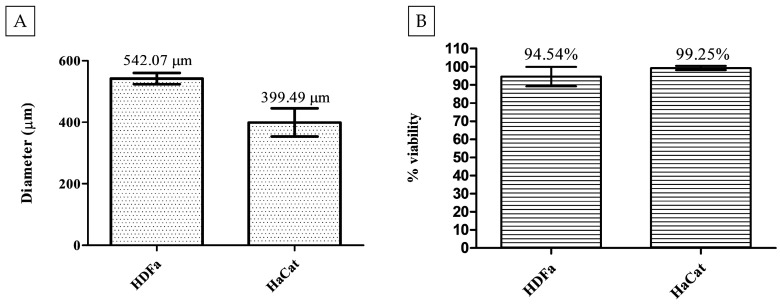
Images illustrate the characterization of the diameter and viability of three-dimensional spheroids. (**A**) The mean diameters of spheroids derived from primary dermal fibroblast (HDFa) and adult human epidermal keratinocyte (HaCat) lines demonstrate their suitability for compound testing. The HDFa cell line formed spheroids with an average diameter of 542.07 ± 18.15 μm, while the HaCat cell line produced spheroids with an average diameter of 399.49 ± 45.53 μm. (**B**) The resazurin method demonstrated high cell viability in both dermal cell lines, with spheroids from the HDFa cell line showing 94.54% viability (standard deviation ± 5.35%) and the HaCat cell line exhibiting 99.25% viability (standard deviation ± 1.14%). The data are presented as mean ± standard deviation.

**Figure 3 pharmaceutics-16-01523-f003:**
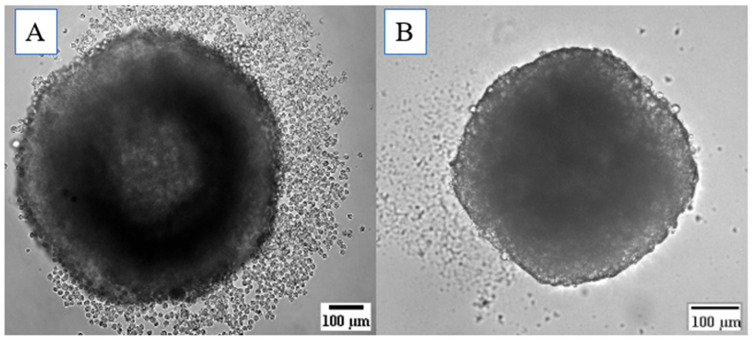
Representative image of a rounded spheroid from the HDFa line, measuring approximately 591.02 µm (**A**). The spheroid from the HaCat line measures 432.36 µm (**B**).

**Figure 4 pharmaceutics-16-01523-f004:**
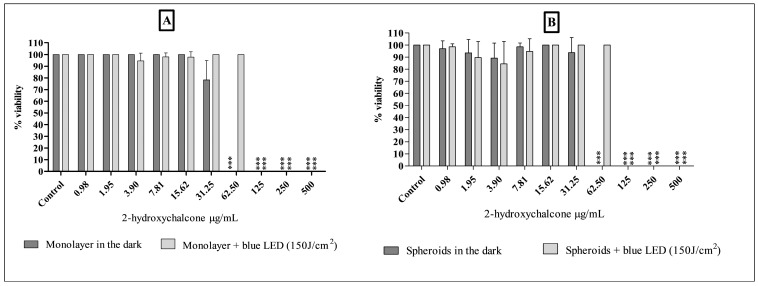
The viability of HDFa cells was evaluated using the resazurin reduction assay following exposure to 2-hydroxychalcone and blue light irradiation at a dose of 150 J/cm^2^ and 2-hydroxychalcone kept in the dark. Results are shown for both the monolayer model (**A**) and the spheroid model (**B**). All values are expressed in micrograms per milliliter (μg/mL). The results of the three independent experiments are expressed as the mean ± standard deviation (SD). The statistical significance of the results was determined using one-way ANOVA, with a *p*-value of less than 0.001 indicated by asterisks (***).

**Figure 5 pharmaceutics-16-01523-f005:**
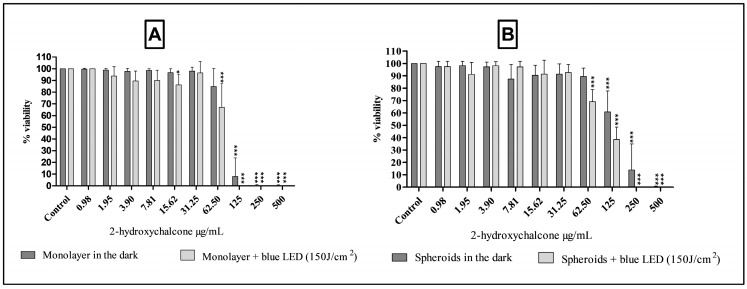
The viability of HaCat cells was evaluated using the resazurin reduction method after exposure to 2-hydroxychalcone and blue light irradiation at a dose of 150 J/cm^2^ and 2-hydroxychalcone kept in the dark. The results are presented for both the monolayer model (**A**) and the spheroid model (**B**). Values are expressed in micrograms per milliliter (μg/mL). The results of the three independent experiments are expressed as the mean ± standard deviation (SD). The statistical significance of the results was determined using one-way ANOVA, with an asterisk (*) indicating a *p*-value of less than 0.05; (***) indicating a *p*-value of less than 0.001.

**Figure 6 pharmaceutics-16-01523-f006:**
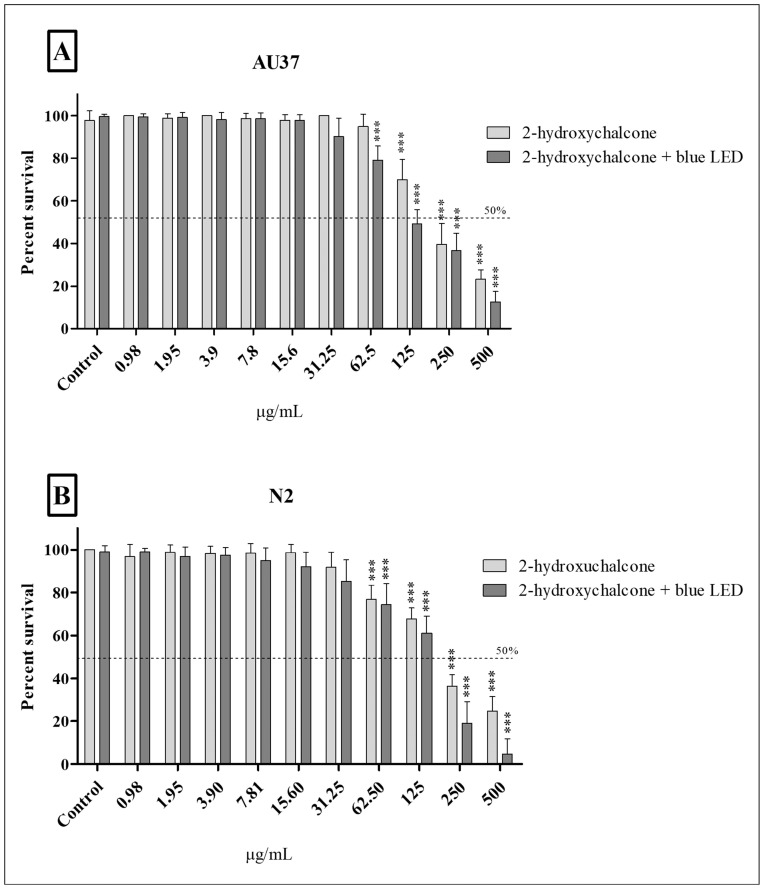
The percentage survival of *Caenorhabditis elegans* at larval stage 4 (L4). Mutant strain AU37 (**A**) and wild-type N2 (**B**) after 24 h of treatment with different concentrations of 2-hydroxychalcone in the dark and 2-hydroxychalcone-mediated PDT. The results of the three independent experiments are expressed as the mean ± standard deviation (SD). The statistical significance of the results was determined using one-way ANOVA, with *** *p* < 0.001.

**Figure 7 pharmaceutics-16-01523-f007:**
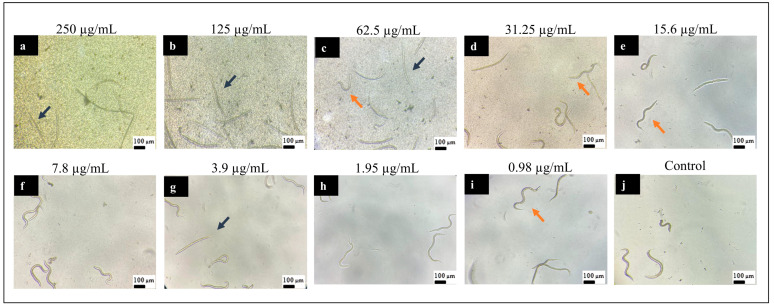
Representative image of *Caenorhabditis elegans* (wild-type strain N2) larvae at the L4 larval stage treated with different concentrations of 2-hydroxychalcone and kept in the dark: 250 µg/mL (**a**); 125 µg/mL (**b**); 62.5 µg/mL (**c**); 31.25 µg/mL (**d**); 15.6 µg/mL (**e**); 7.8 µg/mL (**f**); 3.9 µg/mL (**g**); 1.95 µg/mL (**h**); 0.98 µg/mL (**i**); and untreated control (**j**) (magnification 10×). Rod-shaped larvae were considered dead (blue arrows), while sinusoidal larvae were considered alive (orange arrows).

**Figure 8 pharmaceutics-16-01523-f008:**
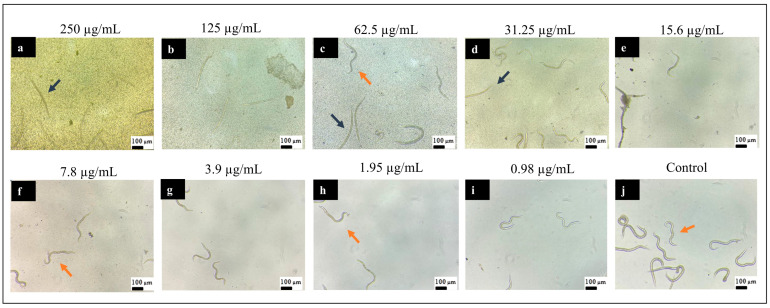
Representative image of *C. elegans* N2 larvae at the L4 larval stage treated with different concentrations of 2-hydroxychalcone mediated by PDT: 250 µg/mL (**a**); 125 µg/mL (**b**); 62.5 µg/mL (**c**); 31.25 µg/mL (**d**); 15.6 µg/mL (**e**); 7.8 µg/mL (**f**); 3.9 µg/mL (**g**); 1.95 µg/mL (**h**); 0.98 µg/mL (**i**); and untreated control (**j**) (magnification 10×). Rod-shaped larvae were considered dead (blue arrows), while sinusoidal larvae were considered alive (orange arrows).

**Figure 9 pharmaceutics-16-01523-f009:**
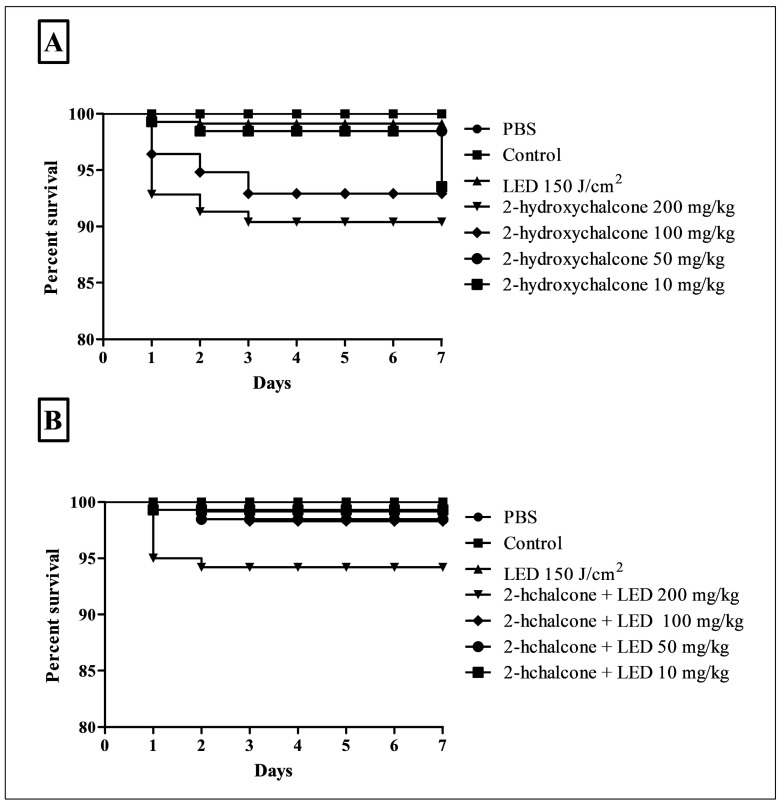
Survival curve of *G. mellonella* larvae treated with different concentrations of 2-hydroxychalcone (10, 50, 100, and 200 mg/kg) in the dark (**A**) and mediated by PDT (**B**). The survival curves were plotted of three independent assays as the mean ± standard deviation (SD).

## Data Availability

The original contributions presented in this study are included in the article/Appendix A. Further inquiries can be directed to the corresponding author.

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
