# Peer review of "Toxicological Assessment of 2-Hydroxychalcone-Mediated Photodynamic Therapy: Comparative In Vitro and In Vivo Approaches"

_pharmaceutics, 2024, doi:10.3390/pharmaceutics16121523_

Round 1
Reviewer 1 Report
Comments and Suggestions for Authors
The article “Effects of Photodynamic Therapy in Alternative In Vitro and In Vivo Model” is devoted to evaluating the impact of photodynamic therapy (PDT) in alternative models to mammals, including in vitro, three-dimensional cultures, and in vivo, in invertebrate animals, utilizing a 2-hydroxychalcone photosensitizer. Based on the contents of the manuscript, the authors have tried to broaden the application of PDT against fungal infections. However, there are several issues that need to be addressed. Based on my review, I suggest a major revision of this manuscript.
Major revision.
1. Introduction section.
In general, the section is weak and should be thoroughly revised. In current form, it does not clearly provide the state-of-the-art of the study. The first paragraph on fungal infections is somewhat confusing to the reader who was expecting an explanation of the need to develop new models for PDT. Various 2D and 3D models in vitro and vivo models already exist. What makes them so unsuitable for PDT? The authors should clearly state their disadvantages and the necessity to create new ones. There is no clear explanation for the author’s interest in 2-hydroxychalcone photosensitizer and its characteristics.
2. Materials and Methods section.
2.2 Please provide several references on main characteristics of 2-hydroxychalcone, if it’s possible – what class of generation? Their spectral characteristics?
“2-hydroxychalcone was added to the cells and incubated for 10 minutes in the dark” – Is this time sufficient for the photosensitizer to penetrate and localize in the cell?
“Cells were irradiated with blue LED at a dose of 150 J. cm-2” – Please specify the time of exposure
3. Results section.
Low quality of figures, especially Fig. 4, 5, 6 – not visible
According to Fig. 5 and 6, light irradiation did not significantly (globally) influence on HaCat and HDFa cells. Please give some explanations\suggestions why, because so far it seems that the time of incubation with the photosensitizer and PDT mode have been incorrectly selected.
In my opinion the authors should provide evidence that the photosensitizer accumulates sufficiently in cells to realize its effects in PDT. This comment concerns both in vitro and in vivo studies.
Fig. 7 and 8 – is it possible to indicate (with arrows) in the images the significant changes to which readers should pay attention? Why are the images somewhat cloudy (with dots) for concentrations up to 15.6 µg/mL?
Author Response
The article “Effects of Photodynamic Therapy in Alternative In Vitro and In Vivo Model” is devoted to evaluating the impact of photodynamic therapy (PDT) in alternative models to mammals, including in vitro, three-dimensional cultures, and in vivo, in invertebrate animals, utilizing a 2-hydroxychalcone photosensitizer. Based on the contents of the manuscript, the authors have tried to broaden the application of PDT against fungal infections. However, there are several issues that need to be addressed. Based on my review, I suggest a major revision of this manuscript.
The authors are grateful for all suggestions and corrections made to the manuscript. All changes are highlighted in yellow.
Major revision.
- Introduction section.
In general, the section is weak and should be thoroughly revised. In current form, it does not clearly provide the state-of-the-art of the study. The first paragraph on fungal infections is somewhat confusing to the reader who was expecting an explanation of the need to develop new models for PDT. Various 2D and 3D models in vitro and vivo models already exist. What makes them so unsuitable for PDT? The authors should clearly state their disadvantages and the necessity to create new ones. There is no clear explanation for the author’s interest in 2-hydroxychalcone photosensitizer and its characteristics.
The introduction section has been improved.
- Materials and Methods section.
2.2 Please provide several references on main characteristics of 2-hydroxychalcone, if it’s possible – what class of generation? Their spectral characteristics?
More references, information and images of the chemical structure and absorption spectrum have been added to the manuscript, as shown in lines 140-147.
“2-hydroxychalcone was added to the cells and incubated for 10 minutes in the dark” – Is this time sufficient for the photosensitizer to penetrate and localize in the cell?
In previous experiments, we have shown that this 10-minute incubation time in the dark for 2-hydroxychalcone is valid since it was able to penetrate the cells of dermatophytes (eukaryotes) and cause damage, either to cells in planktonic form or biofilms (https://doi.org/10.3389/fmicb.2020.01154). In addition, 2-hydroxychalcone is a chalcone derivative modified with two hydroxyl groups, which increases its hydrophilicity and facilitates penetration into the biofilm's extracellular matrix. Despite this modification, its predominant lipophilic characteristics allow efficient permeation through the cell membrane, giving the molecule significant penetration capacity.
“Cells were irradiated with blue LED at a dose of 150 J. cm-2” – Please specify the time of exposure.
Power approx. 116 mW or 0.116 Watts
Time 2586 seconds or 43 minutes
Area approx. = 2 cm 2
Dose = 150 J /cm2
Details of the irradiation time have been added in the Lines – 159-160
- Results section.
Low quality of figures, especially Fig. 4, 5, 6 – not visible
All images have been resampled and have a resolution of 600 dpi.
According to Fig. 5 and 6, light irradiation did not significantly (globally) influence on HaCat and HDFa cells. Please give some explanations\suggestions why, because so far it seems that the time of incubation with the photosensitizer and PDT mode have been incorrectly selected.
The photosensitization-induced photodynamic (PDT) action time was previously determined in studies on the antifungal and antibiofilm activity of Histoplasma capsulatum and dermatophytes (https://doi.org/10.1016/j.pdpdt.2017.03.001; https://doi.org/10.3389/fmicb.2020.01154). In the present study, we continued to utilize the same photosensitizer and incubation time in order to demonstrate that this photosensitizer (2-hydroxyclacone) exhibits greater specificity for fungal cells, both in planktonic form and in biofilm, than for human cells (HaCat and HDFa).
In my opinion the authors should provide evidence that the photosensitizer accumulates sufficiently in cells to realize its effects in PDT. This comment concerns both in vitro and in vivo studies.
The authors understand the concerns raised. However, previous studies with eukaryotic cells, namely H. capsulatum (https://doi.org/10.1016/j.pdpdt.2017.03.001) and dermatophytes (https://doi.org/10.3389/fmicb.2020.01154) biofilms, have already demonstrated the effect of 2-hydroxychalcone mediated PDT with 5-10 min incubation in the dark before irradiation with blue light. As demonstrated for these fungi, the same was observed in this work
Fig. 7 and 8 – is it possible to indicate (with arrows) in the images the significant changes to which readers should pay attention? Why are the images somewhat cloudy (with dots) for concentrations up to 15.6 µg/mL?
Arrows have been added to the images and indicated in the caption to facilitate understanding. The points shown from the highest concentration (250 µg/mL) to the lower concentration (15.6 µg/mL) are typical of the photosensitizer; the higher the concentration, the greater the turbidity of the medium.
Reviewer 2 Report
Comments and Suggestions for Authors
The study on the assessment of PDT using alternative models to mammals provides valuable insights into the effectiveness and safety of this treatment modality. I suggest a Minor revision.
1. The Figure 3 images only show a nanoparticle. This does not fully demonstrate the dispersivity of materials, 'particle aggregation' or 'impurity'? Please explain it.
2. In the early 1900s, German researcher Oscar Raab, under Herman von Tappeiner's supervision, made a key discovery in photodynamic therapy (PDT). He found that exposing Paramecium to acridine and light led to its death. Building on this, von Tappeiner explored using eosin dye with light for carcinoma treatment and coined the term "photodynamic" in 1907. His work established a foundation for further PDT research as a potential cancer treatment. Since then, significant advancements have been made in understanding PDT mechanisms and exploring various photosensitizers (PSs) and light sources for medical applications. In lines 71-74, “Photodynamic therapy has been established in microbiology for over 100 years (DOI: 10.1016/j.phrs.2024.107150; DOI: 10.1111/j.1445-2197.1991.tb00230.x). Photodynamic therapy (PDT) has recently gained attention as a promising antimicrobial alternative, particularly in combating microbial resistance and eradicating biofilms (DOI: 10.3390/nano14151250; DOI: 10.3390/jcm8101581).” Authors are encouraged to reference a selection of pertinent literature.
3. Make sure all abbreviations are written out in full the first time used. This is particularly important in the abstract, such as “Photodynamic therapy (PDT)”.
4. Testing additional invertebrate models or organ-on-chip technologies could further validate the findings and enhance the applicability of alternative models for toxicity testing in PDT research.
5. Conducting comparative studies with other known photosensitizers could help position 2-hydroxychalcone within the broader context of PDT agents and identify its relative advantages or limitations.
6. The chemical structure of 2-hydroxychalcone should be listed.
Comments on the Quality of English LanguageThe English could be improved.
Author Response
The study on the assessment of PDT using alternative models to mammals provides valuable insights into the effectiveness and safety of this treatment modality. I suggest a Minor revision.
- The Figure 3 images only show a nanoparticle. This does not fully demonstrate the dispersivity of materials, 'particle aggregation' or 'impurity'? Please explain it.
The authors are grateful for all comments and suggestions; all changes throughout the manuscript are highlighted in green. Figure 3 shows the three-dimensional culture formed by HDFa and HaCat cells as described in lines 260-267.
- In the early 1900s, German researcher Oscar Raab, under Herman von Tappeiner's supervision, made a key discovery in photodynamic therapy (PDT). He found that exposing Paramecium to acridine and light led to its death. Building on this, von Tappeiner explored using eosin dye with light for carcinoma treatment and coined the term "photodynamic" in 1907. His work established a foundation for further PDT research as a potential cancer treatment. Since then, significant advancements have been made in understanding PDT mechanisms and exploring various photosensitizes (PSs) and light sources for medical applications. In lines 71-74, “Photodynamic therapy has been established in microbiology for over 100 years (DOI: 10.1016/j.phrs.2024.107150; DOI: 10.1111/j.1445-2197.1991.tb00230.x). Photodynamic therapy (PDT) has recently gained attention as a promising antimicrobial alternative, particularly in combating microbial resistance and eradicating biofilms (DOI: 10.3390/nano14151250; DOI: 10.3390/jcm8101581).” Authors are encouraged to reference a selection of pertinent literature.
The authors are grateful for the suggestion. Some references to PDT have been added to the introduction.
- Make sure all abbreviations are written out in full the first time used. This is particularly important in the abstract, such as “Photodynamic therapy (PDT)”.
Thank you for your comment, the changes have been made as indicated in lines 17 and 44.
- Testing additional invertebrate models or organ-on-chip technologies could further validate the findings and enhance the applicability of alternative models for toxicity testing in PDT research.
The authors appreciate the suggestion and agree with the reviewer. Our group is committed to exploring additional invertebrate models and organ-on-a-chip technologies, as well as investigating other 3D culture approaches, to validate these methods as effective alternatives for toxicity testing in photodynamic therapy (PDT) research.
- Conducting comparative studies with other known photosensitizers could help position 2-hydroxychalcone within the broader context of PDT agents and identify its relative advantages or limitations.
We fully agree with the reviewer, future studies are planned to make this comparison in the different photosensitizers. We have already noted that methylene blue is not a good photosensitizer for G. mellonella because its color makes it impossible to verify its melaninization.
- The chemical structure of 2-hydroxychalcone should be listed.
The chemical structure and absorption spectrum have been added in the Materials and Methods section, lines 140-146.
Reviewer 3 Report
Comments and Suggestions for Authors
General comments:
The manuscript lacks important information related to characterization of photosensitizer, illumination experiments, explanation the mechanism of action and results of in-vivo experiments. Introduction on both PDT and selected photosensitizer should be expanded. The title is to general and, in my opinion, Figures 1 and 2 are completely superfluous.
Specific comments:
Line 2: The title is to broad, please specify both effects of PDT and used in vivo and in vitro models.
Line 19 (and 43): please explain in which way are these models safer from mammals’ models?
Line 38: Are fungal infections cause 1.5 miliion deaths per year or they have just have a potential for such damage?
Line 57: As an advantage of using, for example, invertebrates instead of mammals is mentioned the phrase “fewer ethical concerns”. What does it mean? The fate of all used animals is the same. Please remove this phrase from the manuscript. What are the main disadvantages of this approach?
Lines 106-107: Is it really necessary to inform the audience to whom belongs this cell analyser?
Line 108: Which version of the software is used? Software’s producer data are missing.
Section 2.2.2. Characterization, at least optical, of used photosensitizer is missing, for example, a UV/Vis spectra will be useful. Here, the injection in prolegs is used. Please explain how this photosensitizer may be administered into human cancer.
Line 134: There is some problem with number of cells per well, please check.
Line 137: For how long samples were illuminated? Which LED lamp was used? How you measure the light dose? Why only one dose is used?
Lines 149-150: As it is written: some plates were on some plates kept in dark? Please rewrite, if necessary.
Line 181: The form of moving larvae is far away from sinusoidal. Please replace “sinusoidal” with, for example, “wavy”.
Lines 181-182: Which microscope and objective lens are used in experiments?
Line 188: Why it is important that larvae do not have dark spots? Where these spots are placed on larvae?
Line 191: It would be highly beneficial to include several microscopic images of larvae to monitor distribution of injected photosensitizer during observation period.
Lines 195-196: Please be more specific: what kind of touch? How long the larvae should be immobile to be considered dead?
Figures 1 and 2 are not really necessary. It is enough to mention these results in the text.
Please reformulate the title of Section 3.2
Line 258: *** is not an asterisk.
Please use abbreviations for PDT, 2D, and 3D in the whole manuscript.
Mechanism of cell death should be explained, as well as generation and quantification of reactive oxygen species.
References 26 and 39 are the same. Please fix this mistake.
Author Response
General comments:
The manuscript lacks important information related to characterization of photosensitizer, illumination experiments, explanation the mechanism of action and results of in-vivo experiments. Introduction on both PDT and selected photosensitizer should be expanded. The title is to general and, in my opinion, Figures 1 and 2 are completely superfluous.
The authors are grateful for all comments and suggestions. All changes are highlighted in pink in the manuscript.
Specific comments:
Line 2: The title is to broad, please specify both effects of PDT and used in vivo and in vitro models.
The title was modificatior.
Line 19 (and 43): please explain in which way are these models safer from mammals’ models?
The use of in vitro models, such as 3D cell cultures with immortalized cells, has been consolidated as a safer and more ethical alternative to the use of mammals in research. This advance is driven by international regulations, which restrict or prohibit testing on mammals, especially in areas such as cosmetics, where there is a growing demand for cruelty-free products. Not only do 3D models offer a better reproduction of the characteristics of living tissues, but they also meet animal welfare guidelines that promote the use of alternatives whenever possible. When the use of living organisms is necessary, invertebrate models are prioritized due to their lower neural complexity, which minimizes suffering and reduces ethical concerns. In addition, these organisms have other advantages: lower biological risk, reduced costs, and operational ease, in addition to enabling rapid studies due to their short life cycle and ease of genetic manipulation.
Line 38: Are fungal infections cause 1.5 million deaths per year, or they have just have a potential for such damage?
Fungal infections cause more than 1.5 million deaths worldwide each year, according to estimates derived from public health data (World Health Organization (WHO) - WHO fungal priority pathogens list to guide research, development and public health action. Geneva: World Health Organization; 2022) and epidemiological surveys (doi:10.3390/jof3040057; https://doi.org/10.1016/S1473-3099(17)30303-1; 10.1016/S1473-30992300692-8). This number reflects the impact of serious infections that primarily affect immunocompromised individuals, such as patients with HIV/AIDS, cancer, and other health conditions that weaken the immune system. It is important to remember that these statistics are projections, and the actual number of deaths may be underestimated because fungal infections are not notifiable diseases in many countries. In addition, diagnostic errors are common, making it difficult to obtain accurate and complete data on the true extent of these infections.
Line 57: As an advantage of using, for example, invertebrates instead of mammals is mentioned the phrase “fewer ethical concerns”. What does it mean? The fate of all used animals is the same. Please remove this phrase from the manuscript. What are the main disadvantages of this approach?
The phrase "fewer ethical concerns" refers to the fact that the use of invertebrates, such as G. mellonela and C. elegans, in scientific research tends to raise fewer ethical concerns than the use of mammals. This is because invertebrates generally have less complex nervous systems, making them less susceptible to the experience of pain and suffering comparable to that of mammals. As a result, many ethics committees are more flexible in allowing experiments on invertebrates than on mammals. Of course, there are disadvantages to using invertebrates, such as physiological and behavioral differences, limited modeling of human diseases, etc., but these models are in line with the 3Rs and current legislation (https://doi.org/10.1016/j.yrtph.2021.104953; https://doi.org/10.1017/S0007114517002227), and show excellent similarity to mammals, as other studies have shown.
Lines 106-107: Is it really necessary to inform the audience to whom belongs this cell analyser?
Yes, as it is multi-user equipment and has been purchased with public funds, all used equipment belonging to a publicly accessible laboratory must be identified and where it belongs, according to the rules set by the University and the funding bodies.
Line 108: Which version of the software is used? Software’s producer data are missing.
Additional information has been added to the manuscript.
Section 2.2.2. Characterization, at least optical, of used photosensitizer is missing, for example, a UV/Vis spectra will be useful. Here, the injection in prolegs is used. Please explain how this photosensitizer may be administered into human cancer.
More information on the photosensitizer has been added in lines 140-146. Previous studies have shown that 2-hydroxychalcones have antifungal and antibiofilm activity against Histoplasma capsulatum (https://doi.org/10.1016/j.pdpdt.2017.03.001) and dermatophytes (https://doi.org/10.3389/fmicb.2020.01154). The aim of this study was to evaluate the safety profile of chalcones with this therapeutic potential using dermal cell lines. The choice of dermal cells is justified because they represent common sites of infection by skin fungi and provide a relevant model for evaluating the safety and efficacy of chalcones in this context.
Line 134: There is some problem with number of cells per well, please check.
We have checked and the number of cells per well in the manuscript is correct. These cell concentrations per well were used in each lineage mentioned. This is because this type of cultivation requires the addition of specific cell concentrations per well so that the model can be standardized in terms of its diameter and number of cells.
Line 137: For how long samples were illuminated? Which LED lamp was used? How you measure the light dose? Why only one dose is used?
Additional features have been added to the text on lines 157-160. A blue LED (IrradLED®-Biopdi, São Carlos, SP, Brazil) with a wavelength between 455 nm and 492 nm was used as the light source for the samples. The intensity was maintained at 58 mW and the dose administered was 150 J/cm2. A previous study (https://doi.org/10.3389/fmicb.2020.01154) was carried out in which we defined 150 J/cm2 as the ideal dose to effectively inactivate biofilms.
Lines 149-150: As it is written: some plates were on some plates kept in dark? Please rewrite, if necessary.
The sentence has been rewritten for easier comprehension as lines 170-172.
Line 181: The form of moving larvae is far away from sinusoidal. Please replace “sinusoidal” with, for example, “wavy”.
We appreciate the observation. Indeed, the term "sinusoidal" is widely used in the literature to describe the locomotion of the nematode C. elegans, which exhibits a sine-wave-like motion pattern on surfaces such as agar (https://www.nature.com/articles/s41598-021-92690-2; https://doi.org/10.1093/biosci/biu058; https://doi.org/10.1371/journal.pone.0040121). This term is supported by several studies that describe the undulatory locomotion of C. elegans as sinusoidal, which is relevant for biomechanical and behavioral analyses. Therefore, we propose to retain the term "sinusoidal" to ensure consistency with established technical usage in the field.
Lines 181-182: Which microscope and objective lens are used in experiments?
Information has been added to Lines 205-207.
Line 188: Why it is important that larvae do not have dark spots? Where these spots are placed on larvae?
Galleria mellonella larvae produce melanin in response to stressors, which is used in experiments as an indicator of both the quality of the organism and the response of the immune system to potentially toxic compounds. Melanin production can manifest itself as dark spots on the larvae, signaling that they are responding to a stress factor or health threat. It is therefore essential to select larvae without visible spots to ensure the quality of the experimental model and to obtain more reliable results on the toxicity of the compounds being tested. The selection of spotless larvae reduces the interference of pre-existing immune responses, allowing a more accurate assessment of the effects of experimental treatments.
Line 191: It would be highly beneficial to include several microscopic images of larvae to monitor distribution of injected photosensitizer during observation period.
Some images related to the G. mellonella experiments were added over the days for better understanding in supplementary material Figure 1.
Lines 195-196: Please be more specific: what kind of touch? How long the larvae should be immobile to be considered dead?
Galleria mellonella larvae are often quite mobile and death is easy to observe. To confirm the death of the larvae, we perform a gentle touch, but there is no specific time at which a larva is considered dead. The experiment lasts 7 days and the larvae are checked at the same time each day. The absence of any movement, either spontaneous or in response to a gentle touch, is a clear indication that the larva is dead. In addition, other characteristics of dead larvae may be observed, such as melanization of the body, which may be total or partial, resulting in a dark color. It is also common for the larval body to be straight, rather than the typical curvature of a healthy larva. These additional observations help to ensure that the assessment of mortality is accurate and consistent throughout the experiment. As added in Supplementary Material Figure 1.
Figures 1 and 2 are not really necessary. It is enough to mention these results in the text.
Figures 1 and 2 have been combined into Figure 2 (A and B). The aim of demonstrating the characterization of three-dimensional cultures was to simplify the way we perform cytotoxicity experiments and to clarify the methodology. In addition, we wanted to show that the results obtained are only reliable and perishable for replication, including PDT tests, as the 3D model has been little studied for this purpose. In addition, the 3D model shows cell-cell and cell-matrix interactions similar to tissues, making this model more effective for testing new compounds.
Please reformulate the title of Section 3.2 - the section has been reworded.
Line 258: *** is not an asterisk – the “an” was removed.
Please use abbreviations for PDT, 2D, and 3D in the whole manuscript. - The text has been completely revised for abbreviations and they are all highlighted in pink.
Mechanism of cell death should be explained, as well as generation and quantification of reactive oxygen species.
- It was added in the introduction (lines 76-80) about the mechanism of action.
References 26 and 39 are the same. Please fix this mistake.
- Thank you for your comment, the reference has been removed.
Round 2
Reviewer 1 Report
Comments and Suggestions for Authors
All raised issues have been addressed and I don't have any additional comment.
Author Response
We are very grateful to the reviewer
Reviewer 3 Report
Comments and Suggestions for Authors
The authors answered on almost all raised questions and improve the quality of the manuscript which might be considered for publication in Pharmaceutics after following mandatory minor revisions.
Lines 114-115: Usually, such information is announced in the Acknowledgment whereas only information about manufacturer is given in the text. Please do so.
Figure 1B: Replace wavelenght with wavelength on abscissa.
Lines 147-148: The numbers are OK but there is still one “e” between them in “…final concentration of 5 x 102 e 2 x 104 cells per well…”. Please fix this.
Line 164: Put multiplication sign instead of dot after J in “…with blue LED at 150 J. cm-2,…”.
Line 209: Have you ever examined the influence of 70% ethanol on larvae viability?
Line 274: An asterisk is *, *** is not an asterisk but three asterisks.
Figures 4B, 5B and 9B: put 2 in superscript in cm2 (five times in total).
Figure 9: Please change Kg to kg in legend (8 times in total).
Author Response
Lines 114-115: Usually, such information is announced in the Acknowledgment whereas only information about manufacturer is given in the text. Please do so.
Reply: Thank you for your consideration, all changes are highlighted in yellow. The change has been made.
Figure 1B: Replace wavelenght with wavelength on abscissa.
Reply: The axis name has been replaced, thank you.
Lines 147-148: The numbers are OK but there is still one “e” between them in “…final concentration of 5 x 102 e 2 x 104 cells per well…”. Please fix this.
Reply: Sorry, you're right. We thought the problem was with the numbers and didn't notice the letter 'e'. It's fixed now. Thank you very much.
Line 164: Put multiplication sign instead of dot after J in “…with blue LED at 150 J. cm-2,…”.
Reply: We fixed it, thank you.
Line 209: Have you ever examined the influence of 70% ethanol on larvae viability?
Reply: In our study, 70% ethanol was used exclusively for asepsis of G. mellonella larvae, following standard protocols that are widely used. This step was short and aimed only at disinfecting the protopaths. But, it can be observed that this procedure had no effect on the viability of the larvae, since the control groups showed total survival.
Figures 4B, 5B and 9B: put 2 in superscript in cm2 (five times in total).
Reply: All figures have been revised.
Line 274: An asterisk is *, *** is not an asterisk but three asterisks.
Reply: The change has been made.
Figure 9: Please change Kg to kg in legend (8 times in total).
Reply: The change has been made.